# Annotation Sensitivity: Training Data Collection Methods Affect Model Performance

**Christoph Kern**♣   **Stephanie Eckman**◇   **Jacob Beck**♣
**Rob Chew**♠   **Bolei Ma**♣   **Frauke Kreuter**♣,◇

♣LMU Munich   ◇University of Maryland, College Park   ♠RTI International

{christoph.kern, jacob.beck, bolei.ma, frauke.kreuter}@lmu.de
steph@umd.edu   rchew@rti.org

## Abstract

When training data are collected from human annotators, the design of the annotation instrument, the instructions given to annotators, the characteristics of the annotators, and their interactions can impact training data. This study demonstrates that design choices made when creating an annotation instrument also impact the models trained on the resulting annotations.

We introduce the term annotation sensitivity to refer to the impact of annotation data collection methods on the annotations themselves and on downstream model performance and predictions.

We collect annotations of hate speech and offensive language in five experimental conditions of an annotation instrument, randomly assigning annotators to conditions. We then fine-tune BERT models on each of the five resulting datasets and evaluate model performance on a holdout portion of each condition. We find considerable differences between the conditions for 1) the share of hate speech/offensive language annotations, 2) model performance, 3) model predictions, and 4) model learning curves.

Our results emphasize the crucial role played by the annotation instrument which has received little attention in the machine learning literature. We call for additional research into how and why the instrument impacts the annotations to inform the development of best practices in instrument design.

Keywords: Annotation sensitivity, human annotation, annotation instrument, task structure effects

## 1   Introduction

Supervised NLP models are typically trained on human annotated data and assume that these annotations represent an objective "ground truth." Mislabeled data can greatly reduce a model's ability to learn and generalize effectively (Frénay and Verleysen, 2013). Item difficulty (Lalor et al., 2018;

Swayamdipta et al., 2020), the annotation scheme (Northcutt et al., 2021), and annotator characteristics (Geva et al., 2019; Al Kuwatly et al., 2020) can contribute to bias and variability in the annotations. How the prompts are worded, the arrangement of buttons, and other seemingly minor changes to the instrument can also impact the annotations collected (Beck et al., 2022).

In this work, we investigate the impact the annotation collection instrument on downstream model performance. We introduce the term *annotation sensitivity* to refer to the impact of data collection methods on the annotations themselves and on model performance and predictions. We conduct experiments in the context of annotation of hate speech and offensive language in a tweet corpus.

Our results contribute to the growing literature on Data Centric AI (Zha et al., 2023), which finds that larger improvements in model performance often come from improving the quality of the training data rather than model tuning. The results will inform the development of best practices for annotation collection.

## 2   Background and Related Work

Annotation sensitivity effects are predicted by findings in several fields, such as survey methodology and social psychology. Decades of research in the survey methods literature find that the wording and order of questions and response options can change the answers collected (Tourangeau et al., 2000; Schuman and Presser, 1996). For example, small changes in survey questions ("Do you think the government should forbid cigarette advertisements on television?" versus "Do you think the government should allow cigarette advertisements on television?") impact the answers respondents give. Questions about opinions are more sensitive to these effects than questions about facts (Schnell and Kreuter, 2005).

Psychologists and survey methodologists have

also documented the cognitive shortcuts that respondents take to reduce the length and burden of a survey and how these impact data quality. Respondents often satisfice, choosing an answer category that is good enough, rather than expending cognitive effort to evaluate all response options thoroughly (Krosnick et al., 1996). When the questions allow, respondents will go further, giving incorrect answers to reduce the burden of a survey (Kreuter et al., 2011; Tourangeau et al., 2012; Eckman et al., 2014).

Prior research finds strong evidence of task structure effects in annotations of hate speech and offensive language. When annotators coded both annotations on one screen, they found 41.3% of tweets to contain hate speech and 40.0% to contain offensive language. When the task was split over two screens, results differed significantly. The order of the annotations matters as well. When hate speech was asked first, 46.6% of tweets were coded as hate speech and 35.4% as offensive language. The converse was true when the order was switched (39.3% hate speech, 33.8% offensive language) (Beck et al., 2022). Our work expands upon these findings to test the hypothesis that task structure affects not just the distribution of the annotations but also downstream model performance and predictions.

## 3 Methods

We tested our ideas about the downstream effects of annotation task structure in the context of hate speech and offensive language in tweets. We used a corpus of tweets previously annotated for these tasks (Davidson et al., 2017), but replaced the original annotations with new ones collected under five experimental conditions. We then trained separate models on the annotations from each condition to understand how the annotation task structure impacts model performance and predictions.

### 3.1 Data Collection

We sampled tweets from the Davidson et al. corpus, which contains 25,000 English-language tweets. We selected 3,000 tweets, each annotated three times in the prior study. Selection was stratified by the distribution of the previous annotations to ensure that the sample contained varying levels of annotator disagreement. We created nine strata reflecting the number of annotations (0, 1, 2, or 3) of each type (hate speech, offensive language,

neither). The distribution of the sample across the strata is shown in Table A.1 in Appendix §A.1. We do not treat the previous annotations as ground truth and do not use them in our analysis. There is no reason to believe they are more correct than the annotations we collected, and ground truth is difficult to define in hate speech annotation (Arhin et al., 2021).[1]

We developed five experimental conditions which varied the annotation task structure (Figure 1). All tweets were annotated in each condition. Condition A presented the tweet and three options on a single screen: hate speech (HS), offensive language (OL), or neither. Annotators could select one or both of HS and OL, or indicate that neither applied. Conditions B and C split the annotation of a single tweet across two screens (screens are shown in Figure 1 as black boxes). For Condition B, the first screen prompted the annotator to indicate whether the tweet contained HS. On the following screen, they were shown the tweet again and asked whether it contained OL. Condition C was similar to Condition B, but flipped the order of HS and OL for each tweet. Annotators assigned to Condition D first annotated HS for all assigned tweets and then annotated OL for the same set of tweets. Condition E worked the same way but started with the OL annotation task followed by the HS annotation task.

In each condition, annotators first read through a tutorial (Appendix §A.3) which provided definitions of HS and OL (adapted from Davidson et al. 2017) and showed examples of tweets containing HS, OL, both, and neither. Annotators could access these definitions at any time. Annotators were randomly assigned to one of the experimental conditions, which remained fixed throughout the task. The annotation task concluded with the collection of demographic variables and some task-specific questions (e.g., perception of the annotation task, social media use). We assessed demographic balance across experimental conditions and found no evidence of meaningful imbalance (see Figure A.4 in Appendix §A.4).

Each annotator annotated up to 50 tweets. Those in Condition A saw up to 50 screens, and those in other conditions saw up to 100. We collected three annotations per tweet and condition, resulting in 15 annotations per tweet and 44,900 tweet-annotation

---

[1]The original annotation task defined hate speech and offensive language as mutually exclusive categories, which we felt was too restrictive and difficult for annotators.

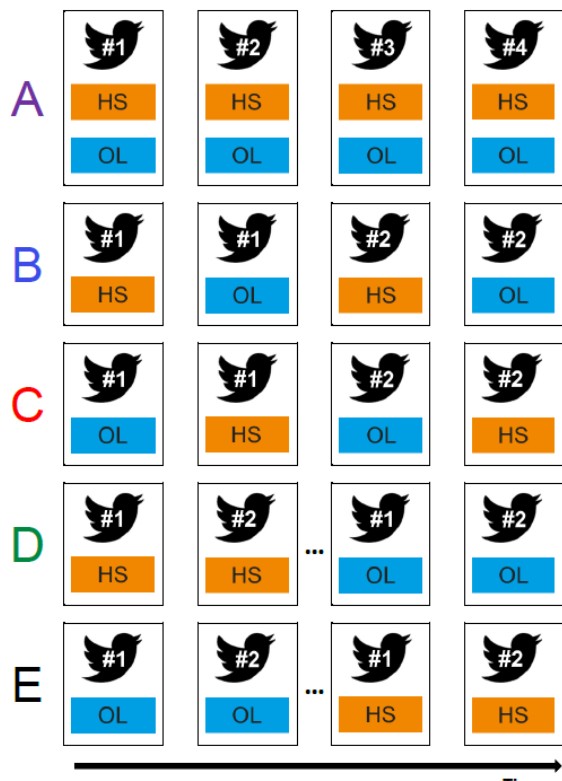

Figure 1: Illustration of Experimental Conditions

combinations.[2]

We recruited 917 annotators via the crowdsourcing platform Prolific during November and December 2022. Only platform members in the US were eligible. Annotators received a fixed hourly wage in excess of the US federal minimum wage after completing the task.

The full dataset including the annotations and the demographic information of the annotators is available at the following HuggingFace repository: https://huggingface.co/datasets/soda-lmu/tweet-annotation-sensitivity-2.

### 3.2 Model Training

**Training and Test Setup**   We used the collected annotations to build prediction models for two binary outcomes: Predicting whether a tweet contains offensive language (1 = yes, 0 = no), and predicting whether a tweet contains hate speech (1 = yes, 0 = no). We split our data on the tweet level into a training set (2,250 tweets) and a test set (750 tweets). Because we collected three annotations per tweet in each experimental condition, we had five condition-specific training data

sets which each contain 6,750 ($2,250 \times 3$) tweet-annotations. We similarly created five condition-specific test sets with 2,250 ($750 \times 3$) tweet-annotations each and a combined test set that contains tweet-annotations from all five conditions. Note that the five condition-specific training sets contain the same tweets, annotated under different experimental conditions, as do the five condition-specific test sets.

We used the five training sets to build condition-specific prediction models for both OL and HS. That is, we trained a model on annotations collected only in Condition A and another model on annotations collected in Condition B, and so on. Models were evaluated against the five condition-specific test sets as well as the combined test set. For both model training and testing, we used the three annotations collected for each tweet in each condition; that is, we did not aggregate annotations to the tweet level (Aroyo and Welty, 2015).

**Model Types**   We used Bidirectional Encoder Representations from Transformers (BERT, Devlin et al. 2019), which is widely used for text classification and OL/HS detection tasks (e.g. Vidgen et al., 2020; Al Kuwatly et al., 2020). BERT is a pre-trained model which we fine-tuned on the annotations from each condition. We also trained Long Short-Term Memory models (LSTM, Hochreiter and Schmidhuber 1997) on the condition-specific training data sets. The LSTM results are reported in Appendix §A.4.

**Training and Validation**   The condition-specific training datasets were further split into a train (80%) and development (20%) set for model validation. During training, after each training epoch, we conducted a model validation on the development set based on accuracy. If the validation showed a higher accuracy score than in the previous epoch, we saved the model checkpoint within this epoch. After full training epochs, we saved the respective best model and used it for final deployment and evaluation on the five condition-specific test sets. We repeated this training process 10 times with different random seeds and report average performance results to make our findings more robust. We report further model training details in Appendix §A.2.

**Reproducibility**   The code for data processing, model training and evaluation, and a list of software packages and libraries used is available at the fol-

---

[2]Annotations by two annotators of 50 tweets each were corrupted and omitted from analysis.

## 4 Results

To understand the influence of the annotation task structure on the annotations themselves, we first compare the percent of tweets annotated as OL and HS and the agreement rates across conditions. We then evaluate several measures of model performance to study the impact of the task structure on the models, including balanced accuracy, ROC-AUC, learning curves, and model predictions. Corresponding results for the LSTM models are given in Appendix §A.4. All statistical test account for the clustering of tweets within annotators.

**Collected Annotations** Table 1 contains the frequency of OL and HS annotations by experimental condition. The number of annotations and annotators was nearly equivalent across the five conditions. More tweets were annotated as OL than HS across all conditions. Condition A, which displayed both HS and OL on one screen and asked annotators to select all that apply, resulted in the lowest percentage of OL and HS annotations. Conditions B and C resulted in equal rates of OL annotations. However, Condition C, which collected HS on the second screen (see Figure 1), yielded a lower share of HS annotations than Condition B ($t = 5.93, p < 0.01$). The highest share of HS annotations was observed in Condition D, which asked annotators to first annotate all tweets as HS and then repeat the process for OL. Condition E, which flipped the task order (first requesting 50 OL annotations followed by 50 HS annotations), resulted in significantly more OL annotations than Condition D ($t = -10.23, p < 0.01$).

| Cond. | Number of | | Percent | |
|---|---|---|---|---|
| | Annotations | Annotators | OL | HS |
| A | 9,000 | 184 | 51.6 | 26.8 |
| B | 9,000 | 183 | 58.8 | 29.6 |
| C | 8,950 | 182 | 58.5 | 28.2 |
| D | 8,950 | 179 | 54.4 | 33.5 |
| E | 9,000 | 189 | 59.0 | 31.8 |

Table 1: Annotation results by condition

Table 2 highlights disagreement across conditions. We calculated the modal annotation for each tweet in each condition from the three annotations, separately for HS and OL, and compared these across conditions. The left table gives the agreement rates for the modal OL annotations and the right table for the modal HS annotations. For OL, Condition A disagrees most often with the other conditions. For HS, the agreement rates are smaller, and Condition E has the most disagreement with the other conditions.

| Cond. | OL | | | | HS | | | |
|---|---|---|---|---|---|---|---|---|
| B | 0.653 | | | | 0.596 | | | |
| C | 0.646 | 0.731 | | | 0.545 | 0.536 | | |
| D | 0.629 | 0.695 | 0.707 | | 0.559 | 0.579 | 0.539 | |
| E | 0.655 | 0.740 | 0.740 | 0.724 | 0.477 | 0.505 | 0.484 | 0.510 |
| | A | B | C | D | A | B | C | D |

Table 2: Agreement between modal labels across annotation conditions (Krippendorff's alpha)

**Prediction Performance** Table 3 shows the balanced accuracy and ROC-AUC metrics for the fine-tuned BERT models when evaluated against the combined test set (with annotations from all conditions). The models achieve higher performance when predicting OL compared to HS, highlighting that HS detection is more difficult (Table 3).

| | Bal. Accuracy | | ROC-AUC | |
|---|---|---|---|---|
| Cond. | OL | HS | OL | HS |
| A | 0.772 | 0.690 | 0.846 | 0.806 |
| B | 0.792 | 0.704 | 0.866 | 0.803 |
| C | 0.802 | 0.681 | 0.862 | 0.800 |
| D | 0.797 | 0.701 | 0.857 | 0.801 |
| E | 0.794 | 0.696 | 0.863 | 0.794 |

Table 3: Balanced Accuracy and ROC-AUC by annotation condition in combined test set, averaged over 10 BERT models

Figure 2 shows the performance results for all combinations of training and testing conditions. The cells on the main diagonal show the balanced accuracy of BERT models fine-tuned on training data from one experimental condition and evaluated on test data from the same condition, averaged across the 10 model runs. The off-diagonal cells show the performance of BERT models fine-tuned on one condition and evaluated on test data from a different condition. The left panel, in blue, contains results from the OL models; the right panel, in orange, from the HS models.

Performance differs across training and testing conditions, particularly for OL, as evident in the row- and column-wise patterns in both panels of

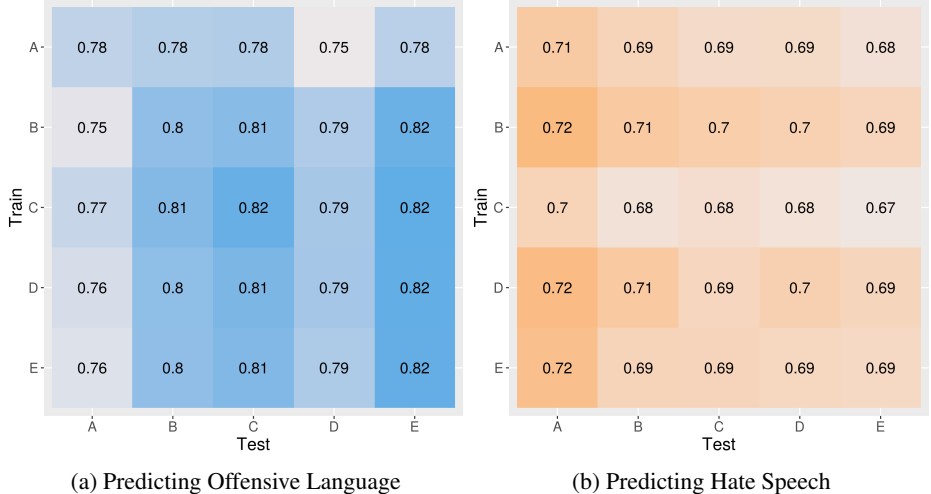

(a) Predicting Offensive Language
(b) Predicting Hate Speech

Figure 2: Performance (balanced accuracy) of BERT models across annotation conditions

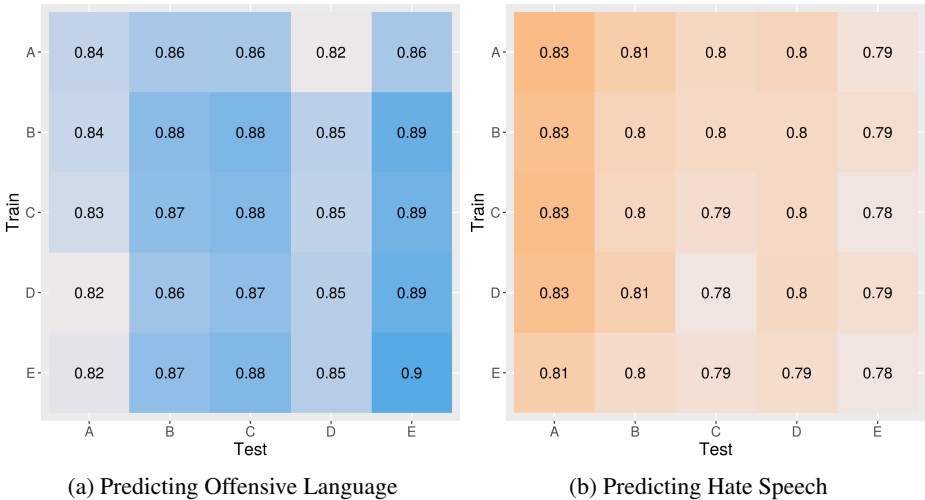

(a) Predicting Offensive Language
(b) Predicting Hate Speech

Figure 3: Performance (ROC-AUC) of BERT models across annotation conditions

Figure 2. We do not observe higher performance scores in the main diagonal: models trained and tested on data from the same experimental condition do not perform better than those trained and tested in different conditions.

Models trained with data from Condition A resulted in the lowest performance across all test sets for the OL outcome (Table 3). Similarly, evaluating OL models on test data from Condition A leads to lower than average performance compared to other test conditions (Figure 2a). These results echo the high disagreement rates in OL annotations between Condition A and the other conditions (Table 2). In contrast, models tested on data from Condition A show the highest performance when predicting HS (Figure 2b).

Conditions B and C collected the HS and OL annotations on separate screens, differing only in the order of the HS and OL screens for each tweet (Figure 1). Models trained on annotations collected in Condition C, where HS was annotated second for each tweet, have lower balanced accuracy across test sets when predicting HS (Figure 2b). However, there is no corresponding effect for Condition B when training models to predict OL.

The OL models perform worse when evaluated on test data collected in Condition D than on data collected in Condition E (see the strong column effects in the last two columns of Figure 2a). Again, no similar column effect is visible in the HS panel (Figure 2b). A potential explanation is that the second batch of 50 annotations was of lower quality

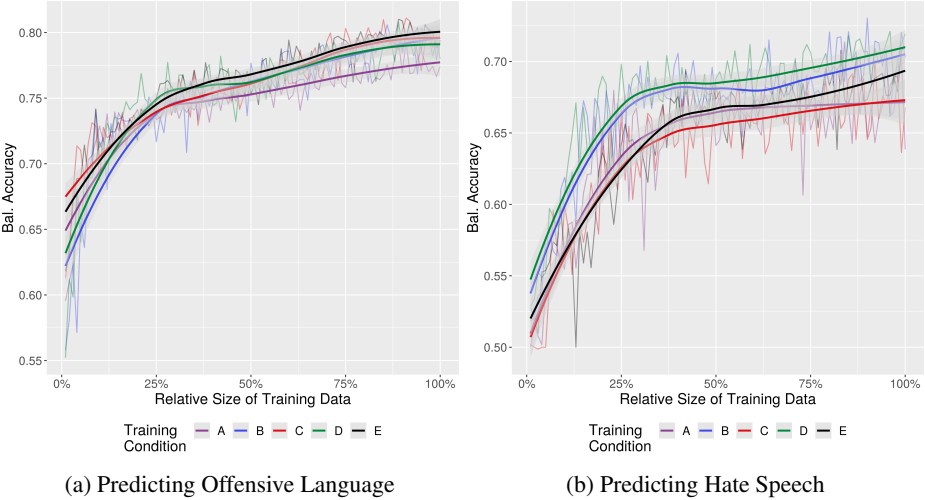

|   | (a) Predicting Offensive Language | (b) Predicting Hate Speech |

(a) Predicting Offensive Language        (b) Predicting Hate Speech

Figure 4: Learning curves of BERT models compared by annotation conditions

due to annotator fatigue.

Evaluating models across conditions with respect to ROC-AUC largely confirms these patterns (Figure 3). The performance differences across testing conditions (column effects) are particularly striking in the ROC-AUC figures, while differences between training conditions (row effects) are less pronounced.

**Predicted Scores**   Comparing model predictions across conditions contextualizes our findings. Table 4 shows agreement (Krippendorf's alpha) between the predictions produced by models fine-tuned on different conditions. All predictions in Table 4 are for tweets in the test sets. OL predictions from the model trained on Condition A show more disagreement with the predictions from the other models (similar to the agreement rates in Table 2 between modal labels). Agreement rates in predictions between models trained on Condition B to E are lower for HS than for OL.

| Cond. | OL | | | | HS | | | |
|---|---|---|---|---|---|---|---|---|
| B | 0.679 | | | | 0.778 | | | |
| C | 0.754 | 0.869 | | | 0.822 | 0.777 | | |
| D | 0.727 | 0.869 | 0.901 | | 0.839 | 0.811 | 0.751 | |
| E | 0.682 | 0.878 | 0.861 | 0.872 | 0.788 | 0.789 | 0.760 | 0.797 |
| | A | B | C | D | A | B | C | D |

Table 4: Agreement between BERT predictions across annotation conditions (Krippendorff's alpha)

**Learning Curves**   We further study how the annotation conditions impact model performance under a range of training set sizes, investigating

whether differences between conditions impact training efficiency. Figure 4 shows learning curves for the BERT models by training condition (line plots and corresponding scatterplot smoothers). These curves show how test set model performance changes as the training data size increases. For each training condition, 1% batches of data were successively added for model training and the model was evaluated on the (combined) test set. To limit computational burden, one model was trained for each training set size and training condition.

While the differences in the learning curves across conditions for OL are small (Figure 4a), Conditions A, C, and E stand out among the HS learning curves (Figure 4b). Building models with annotations collected in Condition E is less efficient: initially, more training data are needed to achieve acceptable performance.

## 5   Discussion

Design choices made when creating annotation instruments impact the models trained on the resulting annotations. In the context of annotating hate speech and offensive language in tweets, we created five experimental conditions of the annotation instrument. These conditions affected the percentage of tweets annotated as hate speech or offensive language as well as model performance, learning curves, and predictions.

Our results underscore the critical role of the annotation instrument in the model development process. Models are sensitive to how training data are collected in ways not previously appreciated in

the literature.

We support the calls for improved transparency and documentation in the collection of annotations (Paullada et al., 2021; Denton et al., 2021). If we assume a universal "ground truth" annotation exists, task structure effects are a cause of annotation noise (Frénay and Verleysen, 2013). Alternatively, if the annotations generated under different task structures are equally valid, task structure effects are a mechanism of researcher-induced concept drift (Gama et al., 2014). Determining a ground truth annotation is difficult and in some instances impossible. In such situations, the documentation of instrument design choices is particularly important. Lack of thorough documentation may explain why attempts to replicate the CIFAR-10 and ImageNet creation process encountered difficulties (Recht et al., 2019).

Large language models (LLMs) can assist with annotation collection, but we expect that human involvement in annotation will continue to be necessary for critical tasks, especially those related to discrimination or fairness. However, such opinion questions are exactly the kinds of questions that are most susceptible to wording and order effects in surveys (Schnell and Kreuter, 2005). Whether LLMs are also vulnerable to task structure effects is not yet known. In addition, we expect that researchers will increasingly use LLMs for pre-annotation and ask annotators review the suggested annotations. This review is vulnerable to anchoring or confirmation bias (Eckman and Kreuter, 2011), where the annotators rely too much on the pre-annotations rather than independently evaluating the task at hand. Task structure effects may impact anchoring bias as well.

We want to give special attention to Condition A, which presents all annotation tasks for a given item on the same screen. We suspect this condition is the most commonly used in practice because it requires fewer clicks and thus less annotator effort. The rate of HS and OL annotations was lowest in Condition A. Models trained on data collected in Condition A had the lowest performance for offensive language and the highest performance for hate speech. There are several possible explanations for the differences in the annotations and in model performance which we cannot disentangle with our data. One explanation is an order effect: the hate speech button was shown first in a list of classes (response options). Respondents may have satisfied by choosing just

one annotation rather than all that apply (Krosnick et al., 1996; Pew Research Center, 2019). Another explanation is distinction bias: asking the annotator to consider both classes simultaneously may lead the annotators to over-examine the distinctions between offensive language and hate speech (Hsee and Zhang, 2004).

Annotators in Conditions D and E annotated 50 tweets and then saw the same tweets again. We hypothesize that the relatively poor performance of these conditions is the result of a fatigue effect. Annotators may give lower-quality annotations in the second half of the task due to boredom and fatigue, which could partially explain the patterns we see in model performance (Figures 2, 3). However, we have not yet tested for order or fatigue effects.

Our results do not clearly demonstrate the superiority of any of the five conditions we tested. More research is needed to identify the best approach. In the meantime, we suggest incorporating variation in the task structure when annotation instruments are created for human annotators, a point also raised by (Recht et al., 2019). Such variation can also help in creating diverse test sets that protect against the test (column) effects we observe (Figures 2, 3). Relying on only one annotation collection condition leads to performance assessments that are subject to annotation sensitivity, that is, distorted by the design of the annotation instrument.

**Limitations** This research has explored annotation sensitivity only in English-language tweets annotated by Prolific panel members living in the United States. The variations in annotations, model performance, and model predictions that we see across conditions could differ in other countries and cultures, especially in the context of hate speech and offensive language. The similar effects in surveys, which motivated our work, do differ across cultures (Tellis and Chandrasekaran, 2010; Lee et al., 2020). This work has also not demonstrated that task structure effects appear in other tasks, such as image and video annotation. In retrospect, we should have included a sixth condition that reversed the display of the label options in Condition A, which would have let us estimate order effects.

We encourage future work in this area to address these limitations.

**Future Work** Our results suggest that there are order and fatigue effects in annotation collection. Future work should conduct experiments to esti-

mate these effects. In surveys, measurement error is more common in later questions (Egleston et al., 2011). Future work could also assess fatigue effects by incorporating paradata (Kreuter, 2013), such as mouse movements (Horwitz et al., 2017) and response times (Galesic and Bosnjak, 2009). These behavioral indicators could contextualize how fatigue impacts task performance and data quality.

In subsequent work, we will explore how the annotators' characteristics influence their judgments and interact with the task structure effects. We also plan to diversify the tasks to include image evaluations and assignments that allow for a more accurate determination of ground truth. These enhancements will provide a more comprehensive understanding of annotation sensitivity. Working with tasks that have (near) ground truth annotations would help the field make progress towards the development of best practices in annotation collection.

## Ethics Statement

In our work, we deal with hate speech and offensive language, which could potentially cause harm (directly or indirectly) to vulnerable social groups. We do not support the views expressed in these hateful posts, we merely venture to analyze this online phenomenon and to study the annotation sensitivity.

The collection of annotations and annotator characteristics was reviewed by the IRB of RTI International. Annotators were paid a wage in excess of the US federal minimum wage.

## Acknowledgments

This research received funding support from RTI International and BERD@NFDI.

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

# A Appendix

## A.1 Selection of Tweets

We selected 3,000 tweets from 25,000, stratified by the previous annotations (Davidson et al., 2017). The nine strata and the sample size in each is shown in Table A.1.

| # of Annotations | | | Sample Size | Percent |
|---|---|---|---|---|
| HS | OL | Neither | | |
| 0 | 0 | 3 | 417 | 14 |
| 0 | 1 | 2 | 417 | 14 |
| 0 | 2 | 1 | 417 | 14 |
| 0 | 3 | 0 | 417 | 14 |
| 1 | 0 | 2 | 160 | 5 |
| 1 | 2 | 0 | 417 | 14 |
| 2 | 0 | 1 | 97 | 3 |
| 2 | 1 | 0 | 417 | 14 |
| 3 | 0 | 0 | 241 | 8 |

Table A.1: Distribution of sampled tweets by stratum

We do not weight for the probability of selection in our analysis, because our goal is not to make inferences to the Davidson et al. corpus, which is itself not a random sample of tweets.

## A.2 Model Training Details

Our implementation of BERT and LSTM models was based on the libraries pytorch (Paszke et al., 2019) and transformers (Wolf et al., 2020). During training, we used the same hyperparameter settings of the respective LSTM and BERT models for our 5 different training conditions to keep these variables consistent for comparison purposes. We report the hyperparameter settings of the models in Table A.2 and A.3. The number of parameters for pre-trained BERT (base) is $\sim$ 110M, and the number of parameters for LSTM is 7 025 153. To avoid random effects on training, we trained each model variation with 10 different random seeds $\{10, 42, 84, 420, 567, 888, 1100, 1234, 5566, 7890\}$ and took the average across the models.

All the experiments were conducted on an NVIDIA® A100 80 GB RAM GPU. Within this computation infrastructure, the LSTM model takes

| Hyperparameter | Value |
|---|---|
| encoder | bert-base-cased |
| epochs_trained | 20 |
| learning_rate | $5e^{-5}$ |
| layer_norm_eps | $1e^{-12}$ |
| batch_size | 64 |
| optimizer | AdamW |

Table A.2: Hyperparameter settings of BERT models

| Hyperparameter | Value |
|---|---|
| epochs_trained | 20 |
| learning_rate | $5e^{-5}$ |
| batch_size | 64 |
| embedding_dim | 512 |
| lstm_hidden_size | 512 |
| linear_layer_size | 256 |
| dropout_rate | 0.3 |
| vocab_size | 5 000 |
| bidirectional | True |
| loss | BCELoss |
| | (binary cross entropy) |
| optimizer | Adam |

Table A.3: Hyperparameter settings of LSTM models

approximately 40 seconds per training data condition, and the BERT model takes approximately 15 minutes.

## A.3 Annotator Instructions

You will see a series of tweets. For each tweet, we want you to code if it contains hate speech or contains offensive language. The next few screens explain what hate speech and offensive language are, to help with the coding.

We define hate speech as:

Language that is used to expresses hatred towards a targeted group or language intended to be derogatory, to humiliate, or to insult the members of the group.

Here are examples of tweets that contain hate speech, according to our definition:

"You are a jiggaboo...!"

"they're working on a bill to prevent retards from voting. who knew retards COULD vote? things are starting to make sense now."

"Every slant in #LA should be deported. Those scum have no right to be here. Chinatown should be bulldozed,"

Remember, hate speech is: language that is used to express hatred towards a targeted group or lan-

guage intended to be derogatory, to humiliate, or to insult the members of the group.

However, context matters. If a tweet discusses someone else's hate speech, it may use hateful terms, but the tweet is not hate speech. For example,

"Why no boycott of the racist "redskins"? #Redskins @ChangeTheName"

This tweet contains a hateful term, but the tweeter is making an anti-racist statement. We would label tweet as containing offensive language.

We define offensive language as:

Language that is highly offensive to certain individuals or groups but does not meet the requirements of hate speech.

"Guess who just got an apartment in downtown Columbus? That's right bitch you guessed it, this guy."

This tweet contains offensive language, but it is not hate speech. We would label it as offensive language.

Some tweets contains both hate speech AND offensive language.

"Subtweet me one more time, you dirty chink whore"

Some tweets do not meet our definition of hate speech or offensive language.

"Great lead battle and then Ricky hits Danica for a yellow. Oh boy. #NASCAR"

## A.4 Additional Results

We provide additional results of LSTM models in Table A.4 and Figure A.1, A.2, A.3, as well as comparisons of demographic covariates across conditions in Figure A.4.

| Cond. | Bal. Accuracy | | ROC-AUC | |
|---|---|---|---|---|
| | OL | HS | OL | HS |
| A | 0.846 | 0.806 | 0.747 | 0.637 |
| B | 0.866 | 0.803 | 0.755 | 0.654 |
| C | 0.862 | 0.800 | 0.759 | 0.625 |
| D | 0.857 | 0.801 | 0.734 | 0.655 |
| E | 0.863 | 0.794 | 0.754 | 0.638 |

Table A.4: Balanced Accuracy and ROC-AUC by annotation condition in combined test set, averaged over 10 LSTM models

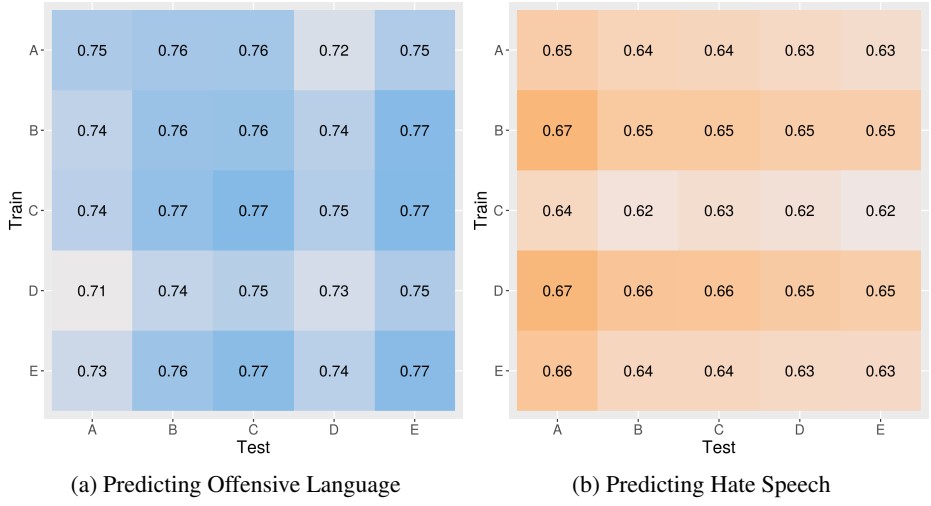

(a) Predicting Offensive Language      (b) Predicting Hate Speech

Figure A.1: Performance (balanced accuracy) of LSTM models across conditions

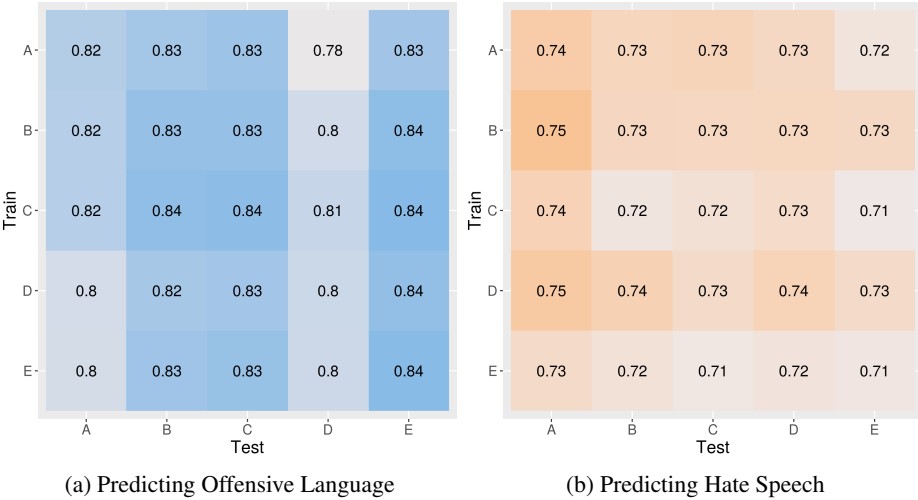

(a) Predicting Offensive Language      (b) Predicting Hate Speech

Figure A.2: Performance (ROC-AUC) of LSTM models across annotation conditions

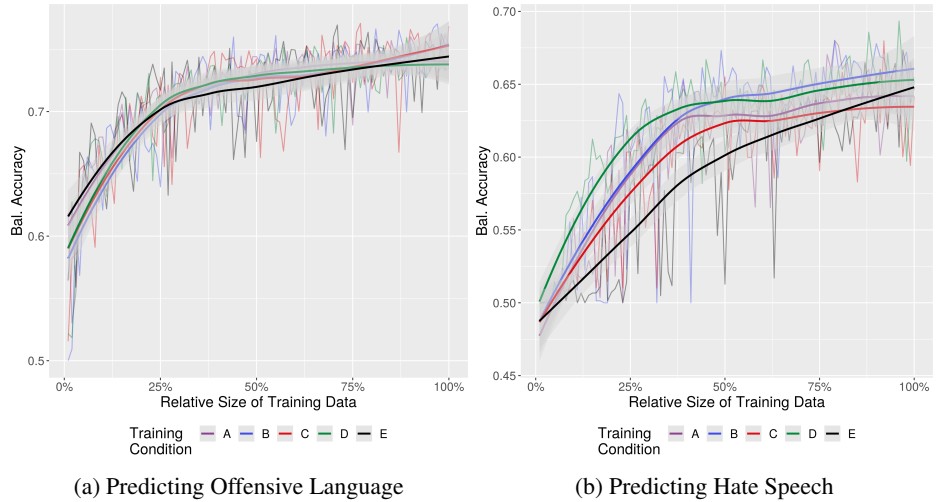

(a) Predicting Offensive Language      (b) Predicting Hate Speech

Figure A.3: Learning curves of LSTM models compared by annotation conditions

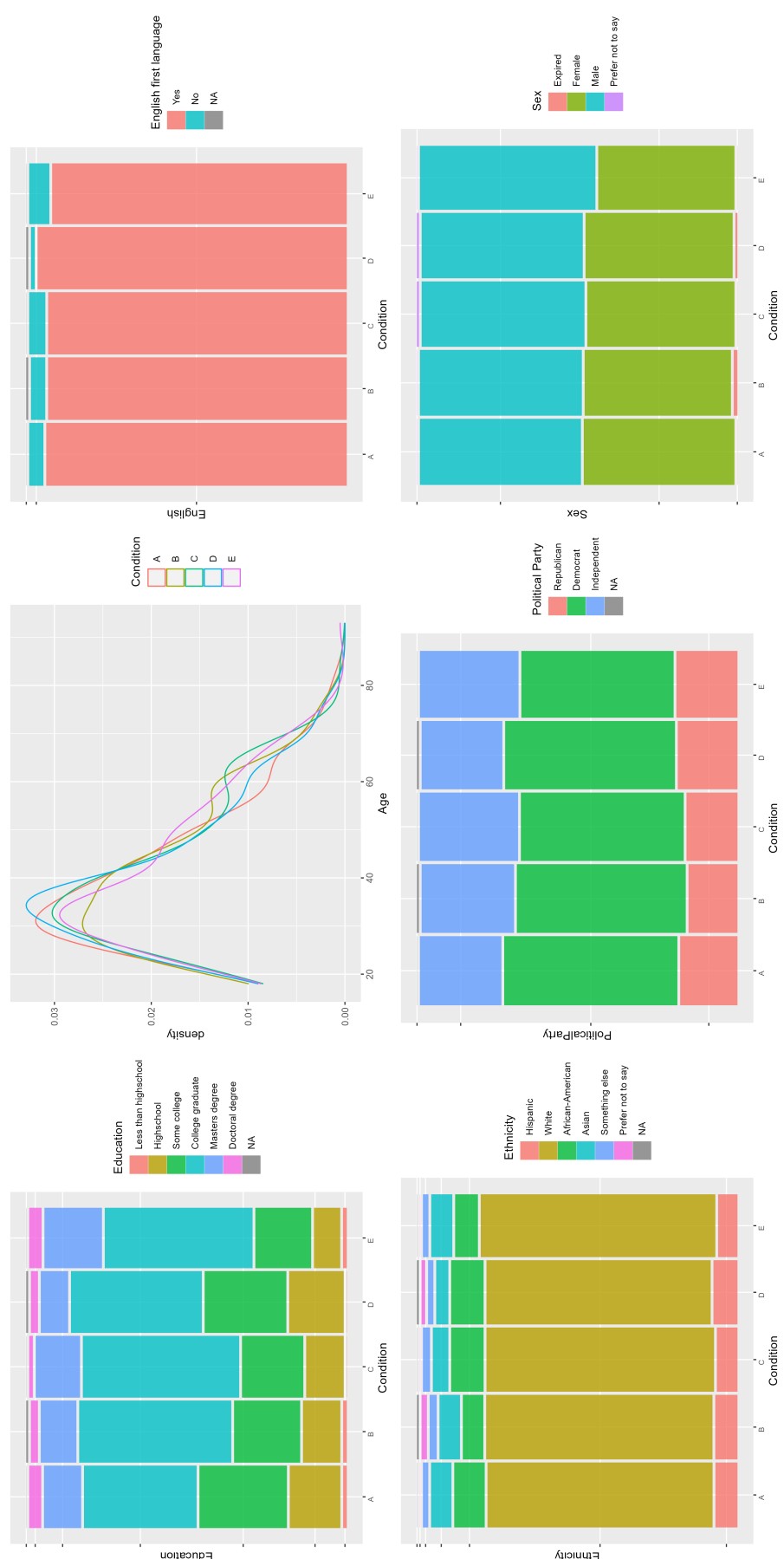

Figure A.4: Comparison of demographic covariates across conditions