# OpenReview forum: "Annotation Sensitivity: Training Data Collection Methods Affect Model Performance"
_EMNLP/2023/Conference — EMNLP 2023 Findings_

### Official Review · Reviewer_W7XF · 2023-08-01

**Soundness:** 4

**Excitement:**

4: Strong: This paper deepens the understanding of some phenomenon or lowers the barriers to an existing research direction.

**Paper Topic And Main Contributions:**

The paper shows how order of annotation questions may affect hate speech (HS)/offensive language (OL) detection. The authors compare five conditions (the conditions are as follows: both labels together, OS followed by HL one tweet at a time, HL followed by OS one tweet at a time, OS of all tweets followed by HL of all tweets, HL of all tweets followed by OS of all tweets).

The analysis/evaluation shows that: (a) Proportions of OS and HL may differ, (b) there may be discrepancies in the HS labels based on the conditions, (c) performance of BERT/LSTM-based classifiers on the same annotation strategy is shown, (d) cross-condition evaluation results may be different, and (e) additional evaluation on prediction scores and training set sizes.

The paper is an interesting study of annotation order, in the context of hate speech detection.

**Questions For The Authors:**

Minor questions (addressing them in the paper will help fill these gaps):
1) Did annotators have the option to abstain from annotating because it was a difficult decision?
2)

**Reasons To Accept:**

1) The study demonstrating the five conditions is examined in the context of the resulting dataset and classifiers trained on the dataset. It makes an interesting read.

2) The observations in the paper open up interesting ideas for future exploration:
(a) Why is hate speech influenced by the annotation condition more than offensive language?
(b) Obtaining multiple labels in one screen is indeed a common annotation strategy, as mentioned in the paper. What does this mean for annotation effort and costs associated with the creation of large annotated datasets?
(c) Are the observations in the paper an artifact of the research problem (OL/HS detection) or does it extend to other tasks?

3) I skimmed through the code included in the link in the paper. The code looks reasonably reproducible.

**Reasons To Reject:**

These are minor reasons, in my opinion. However,
(a) The claims in the paper seem to be exaggerated. The paper mentions 'annotation instrument'. However, the paper describes only one aspect of the annotation instrument.
(b) Annotation sensitivity gives the impression that it is an empirical metric. However, it is a qualitative notion presented in the paper.
(c) While the paper reports many interesting results (as described in the 'contribution' section above), it is a report of an empirical study - without sufficient investigation into the reason for the observation.

**Reproducibility:**

4: Could mostly reproduce the results, but there may be some variation because of sample variance or minor variations in their interpretation of the protocol or method.

**Reviewer Confidence:**

3: Pretty sure, but there's a chance I missed something. Although I have a good feel for this area in general, I did not carefully check the paper's details, e.g., the math, experimental design, or novelty.

**Typos Grammar Style And Presentation Improvements:**

Please include the language of the dataset in the paper. It is neither obvious nor default that English is the language of the dataset.

---

> ### Author Rebuttal · Authors · 2023-08-28
>
> **Response to Review:**
>
> Thank you for your careful reading of our paper.
>
> We apologize for not mentioning the language of the data set. It is indeed English, which we will clarify in the revised paper. As mentioned in the response to Reviewer 1, we will upload the data set and the instructions when we revise the paper.
>
> You are right that we manipulated only one aspect of the annotation instrument. There are of course many design choices that go into an annotation instrument, such as font, color, vertical or horizontal display of the annotation options, button size and type, etc. We chose to focus on the order of the screens and the annotation options to isolate their influence. In future work we will experiment with other aspects of the instrument.
>
> We coined the term annotation sensitivity to draw attention to the fact that the annotations we collect are influenced by (sensitive to) many factors. The term is similar to ``gluten sensitivity'' and other terms. Because this is the first paper where we (or anyone) are using this term, we are open to alternatives.
>
> We agree that our results do not address the mechanisms behind the differences that we observed. Because the influence of minor changes to the annotation instrument design on annotations and models are not well known in the literature, we chose to first demonstrate that these effects occur in this paper. The paper does suggest that a fatigue effect may be at work, but additional research is needed to test for that effect, which we are planning to do.
>
> **Response to Questions For The Authors:**
>
> Earlier research (Beck et al 2022) experimented with including and excluding an option to allow annotators to respond ``Don't Know.'' They found that the option was seldom used (maximum of 2.5\% of responses) and did not change the distribution of the annotations. Because inclusion of this option is not standard in annotation instruments we have seen, we did not include it in this experiment. We will include this information in the revision of the paper.

---

### Official Review · Reviewer_JnQu · 2023-08-05

**Soundness:** 3

**Excitement:**

4: Strong: This paper deepens the understanding of some phenomenon or lowers the barriers to an existing research direction.

**Paper Topic And Main Contributions:**

This paper investigates annotation sensitivity in machine learning models by examining the influence of factors such as the design of annotation instruments, instructions given to annotators, characteristics of annotators, and their interactions on the quality and variability of human-annotated data. The authors focus on annotating hate speech and offensive language in Twitter data, using five experimental conditions of the annotation instrument, each varying in task structure, order, and presentation of annotation options. The study demonstrates that these experimental conditions affect not only the percentage of tweets annotated as hate speech or offensive language, but also model performance in terms of balanced accuracy, ROC-AUC metrics, learning curves, and predictions. The results emphasize the crucial role played by the annotation instrument, which has received little attention in machine learning literature. The authors recommend increased transparency and documentation in the collection of annotations, as well as further research into how and why the collection instrument impacts the annotations collected. This will help support the development of best practices in instrument design. Additionally, the paper suggests incorporating variation in the task structure when creating annotation instruments for human annotators to improve performance assessments and reduce test effects caused by annotation sensitivity.

**Reasons To Accept:**

This paper should be accepted, but as a short paper, because it addresses the crucial issue of annotation sensitivity in machine learning training datasets, which has received little attention in the literature.  The strengths of this paper are:
1- The paper addresses an underexplored topic, annotation sensitivity, which has significant implications for the quality and reliability of machine learning training datasets. By investigating the impact of annotation instrument design on the resulting annotations and model performance, the paper contributes valuable insights to an area that has not received adequate attention in the literature.
2- The paper demonstrates a well-designed experiment with five different annotation conditions, allowing for a comprehensive analysis of the effects of task structure on the collected annotations and the performance of the resulting models. This rigorous approach provides a strong foundation for the findings and their implications.
3- The paper highlights the need for improved transparency, documentation, and best practices in the collection of annotations. These practical implications are of great importance to the NLP community, as they emphasize the need for more robust and reliable training datasets to improve the performance and generalizability of machine learning models.
4- Although the paper focuses on the annotation of hate speech and offensive language in Twitter data, the findings have broader implications for other NLP tasks and domains. The insights gained from this study can inform the design of annotation instruments and data collection strategies in various areas of NLP research.
5- The paper serves as a call to action for the NLP community to further investigate the impact of annotation sensitivity and develop best practices in instrument design. By highlighting the importance of this issue and its potential consequences for machine learning model performance, the paper encourages researchers to explore this topic further and contribute to the development of more reliable and effective NLP techniques.

**Reasons To Reject:**

The paper under review presents an interesting topic regarding the impact of annotation task structure on collected annotations and the performance of machine learning models. However, it might be a good candidate for a short paper publication.
1- The paper does not provide a clear and detailed explanation of the five experimental conditions used in the study. It is important for the authors to elaborate on the reasons behind their choice of conditions and how they differ from each other. This would help readers understand the implications of the findings and their relevance to the research question.
2- The paper presents various findings regarding the impact of task structure on annotations and model performance, but it does not delve deep enough into the reasons behind these differences such as order effects, distinction bias, or fatigue effects. The authors should conduct a more thorough analysis of the results, exploring possible explanations for the observed trends.
3- The paper fails to demonstrate the superiority of any of the five tested conditions in collecting annotations. Without clear evidence supporting one approach over others, it is difficult for readers to draw actionable conclusions from the study. The authors should consider refining their experimental design and analysis to provide more conclusive results.
Despite these limitations, the paper could be a good short paper as it highlights the importance of annotation sensitivity and the crucial role of the annotation instrument in machine learning. It calls for additional research into how and why the collection instrument impacts the annotations collected, which could inform the development of best practices in instrument design. The paper also emphasizes the need for improved transparency and documentation in the collection of annotations and raises interesting questions for future research.


**Reproducibility:**

5: Could easily reproduce the results.

**Reviewer Confidence:**

5: Positive that my evaluation is correct. I read the paper very carefully and I am very familiar with related work.

---

> ### Author Rebuttal · Authors · 2023-08-28
>
> **Response to Review:**
>
> We would like to sincerely thank the reviewer for the thoughtful review. Please see below our responses to your comments -- thank you again for the opportunity to improve the scholarship of this work.
>
> We agree that additional explanation of the experimental conditions will help readers  better interpret the results. We've drafted additional explanations of the experimental conditions for the Appendix. Additionally, we will publish more information of the annotation instrument, as well as the underlying data set, on OSF to help other researchers more easily reproduce our results or extend our work.
>
> We agree that there are several potential mechanisms that are causing the differences we observe. However, in order to attribute the differences to any single (or combination) of these factors, we believe that a follow-up experiment would need to be designed and conducted to test for these specific hypotheses. For example, in the current experiment for Condition A, hate speech was presented as first option in a list of classes (alongside options for "offensive language" and "neither" on the same screen). To be able to disentangle the contributions of order effects vs. distinction bias, we would ideally also need a version of Condition A that presents "offensive language" as the first option to quantify the order effect. We will highlight the need for follow-up attribution studies in Future Work and note this in the Limitations section.
>
> We recognize that our study does not identify a best approach for collection of hate speech and offensive language labels, which is a limitation of the current study. Our long-term goal is to develop actionable best practices for annotation collection. We hope that this early work motivates the broader research community to study this phenomenon by bringing attention to it. However, the revised paper will dedicate more space to discussing potential mechanisms and experimental designs to motivate future work.

---

### Official Review · Reviewer_gqvR · 2023-08-05

**Soundness:** 2

**Excitement:**

2: Mediocre: This paper makes marginal contributions (vs non-contemporaneous work), so I would rather not see it in the conference.

**Paper Topic And Main Contributions:**

This paper dials in on how experiment design decisions, such as the layout of the questions and order of the options, can affect how crowdsourcing workers label instances. They collect 5 sets of annotations using different problem layouts and observe how they influence the performance of their BERT and LSTM models.

**Questions For The Authors:**

A. What was the reasoning for using LSTMs instead of something newer (since it can be predicted that BERT would perform better)?

B. Will you be uploading your data along with the instructions you gave your crowdsourcing workers? That would definitely be another contribution.



**Reasons To Accept:**

The annotations variations caused by differences in MTurk HIT designs are of interest to every and any field that utilizes crowdsourced workers and there is a strong need for understanding how these variations occur.

**Reasons To Reject:**

I'm not sure about the novelty and thoroughness of the project, given that the authors use BERT and LSTMs as their primary models and do not offer metrics more complicated than percentage or accuracy. For any annotation-related study, I'd like to see disagreement metrics, such as Krippendorff's alpha, but that wasn't provided in this paper.

**Reproducibility:**

5: Could easily reproduce the results.

**Reviewer Confidence:**

4: Quite sure. I tried to check the important points carefully. It's unlikely, though conceivable, that I missed something that should affect my ratings.

**Typos Grammar Style And Presentation Improvements:**

Figure 3 doesn't offer much insight despite the amount of space it takes. I would put it in the appendix.

Change the contents of Table 2 from percentage to some disagreement metric. Percentages aren't great for showing how two sets of annotations differ since they also include differences/similarities by chance (For example, two people annotating a binary dataset would most likely have more agreements than two people annotating a dataset with 10 labels)

---

> ### Author Rebuttal · Authors · 2023-08-28
>
> **Response to Review:**
>
> Thank you for taking the time and effort to review our manuscript. We appreciate the feedback provided and will respond to the specific points below.
>
> The focus of our paper is to put forward a clean experimental setup demonstrating the downstream effects of variation in the design of the annotation instrument. In this context, our focus is primarily on *relative* performance differences across both training and evaluation conditions, given common model types and performance metrics that have been used for the prediction task at hand (e.g. Vidgen et al., 2020; Al Kuwatly et al., 2020). While we will update the disagreement metrics (see below), we also welcome further extensions to our work, and thus make the data from our annotation experiment available for the community.
>
> We agree that Krippendorff's alpha is a superior metric of agreement. We will replace the calculations in Table 2 with the Krippendorff's alphas. Our updated analysis shows that the alpha scores largely align with our initial assessment of disagreement patterns across annotation conditions (see Table 1 and 2 below). We will also replace Figure 3 with a table that more concisely summarizes disagreement between the model predictions using Krippendorff's alpha (see Table 3 and 4).
>
> **Response to Questions For The Authors:**
>
> - A. Thank you for pointing this out. We agree that BERT models are more commonly used. We included the LSTM models for robustness and in reference to prior work. In the revised paper, we will move all discussion of the LSTM model results to the appendix.
>
> - B. An important point. As noted above, we will publish the data and a concise documentation file with the manuscript. As a first step, we already updated the [OSF repository](https://osf.io/mn9ux/?view_only=75c84803b70947cb9831bd897cf8f01e) and included the main data files that we used for our analysis. We will also include the instructions given to the annotator in an appendix.
>
> Table 1: Agreement between modal labels for offensive language across annotation conditions (Krippendorff's alpha)
>
> |   | A | B  | C  | D  |
> |---|---|---|---|---|
> | B | 0.653 |   |   |   |
> | C | 0.646 | 0.731 |   |   |
> | D | 0.629 | 0.695 | 0.707 |   |
> | E | 0.655 | 0.74 | 0.74 | 0.724 |
>
> Table 2: Agreement between modal labels for hate speech across annotation conditions (Krippendorff's alpha)
>
> |   | A | B  | C  | D  |
> |---|---|---|---|---|
> | B | 0.596 |   |   |   |
> | C | 0.545 | 0.536 |   |   |
> | D | 0.559 | 0.579 | 0.539 |   |
> | E | 0.477 | 0.505 | 0.484 | 0.51 |
>
> Table 3: Agreement between BERT predictions for offensive language across annotation conditions (Krippendorff's alpha)
>
> |   | A | B  | C  | D  |
> |---|---|---|---|---|
> | B | 0.679 |   |   |   |
> | C | 0.754 | 0.869 |   |   |
> | D | 0.727 | 0.869 | 0.901 |   |
> | E | 0.682 | 0.878 | 0.861 | 0.872 |
>
> Table 4: Agreement between BERT predictions for hate speech across annotation conditions (Krippendorff's alpha)
>
> |   | A | B  | C  | D  |
> |---|---|---|---|---|
> | B | 0.778 |   |   |   |
> | C | 0.822 | 0.777 |   |   |
> | D | 0.839 | 0.811 | 0.751 |   |
> | E | 0.788 | 0.789 | 0.76 |  0.797 |

---

### Meta-Review · Area_Chair_vtFQ · 2023-09-19

**Recommendation:** 3

**Metareview:**

The paper investigates into "annotator sensitivity", i.e., how the wording of instructions affect the annotations collected.

All reviewers agreed this is an interesting and novel topic to look into, but also all raised some concerns on study design and evaluation. In specific:
1. The use of models are not up-to-date
2. The evaluation metric used seem a bit shallow (focusing on model accuracy but not other asspects like annotator-agreements)
3. The investigation focuses on only one type of intervention which makes the claim of "annotator sensitivity" a bit too broad.
4. Multiple artifacts may affect the result, noise of the annotations.

Personally, I find the topic refreshing, and while the study is not perfect, it is sound enough and will spark interest and discussion among NLP researchers. As such, I recommend accepting to Findings track, but would recommend the author to significantly revise the paper as they suggest in the rebuttal.

I would also like to point out a relevant work on instruction bias:
Parmar, Mihir, et al. "Don't Blame the Annotator: Bias Already Starts in the Annotation Instructions." EACL 2023

---

### Decision · Program_Chairs · 2023-10-07

**Decision:**

Accept-Findings

**Comment:**

The paper investigates into "annotator sensitivity", i.e., how the wording of instructions affect the annotations collected.

All reviewers agreed this is an interesting and novel topic to look into, but also all raised some concerns on study design and evaluation. In specific:
1. The use of models are not up-to-date
2. The evaluation metric used seem a bit shallow (focusing on model accuracy but not other asspects like annotator-agreements)
3. The investigation focuses on only one type of intervention which makes the claim of "annotator sensitivity" a bit too broad.
4. Multiple artifacts may affect the result, noise of the annotations.

Personally, I find the topic refreshing, and while the study is not perfect, it is sound enough and will spark interest and discussion among NLP researchers. As such, I recommend accepting to Findings track, but would recommend the author to significantly revise the paper as they suggest in the rebuttal.

I would also like to point out a relevant work on instruction bias:
Parmar, Mihir, et al. "Don't Blame the Annotator: Bias Already Starts in the Annotation Instructions." EACL 2023